# Noninvasive vs invasive respiratory support for patients with acute hypoxemic respiratory failure

Jarrod M. Mosier[1,2]*, Vignesh Subbian[3,4,5], Sarah Pungitore[6], Devashri Prabhudesai[5,7], Patrick Essay[3], Edward J. Bedrick[5,7], Jacqueline C. Stocking[8], Julia M. Fisher[4,5,7]

**1** Department of Emergency Medicine, The University of Arizona College of Medicine, Tucson, Arizona, United States of America, **2** Division of Pulmonary, Allergy, Critical Care, and Sleep, Department of Medicine, The University of Arizona College of Medicine, Tucson, Arizona, United States of America, **3** Department of Systems and Industrial Engineering, College of Engineering, The University of Arizona, Tucson, Arizona, United States of America, **4** Department of Biomedical Engineering, College of Engineering, The University of Arizona, Tucson, Arizona, United States of America, **5** BIO5 Institute, The University of Arizona, Tucson, Arizona, United States of America, **6** Program in Applied Mathematics, The University of Arizona, Tucson, Arizona, United States of America, **7** Statistics Consulting Laboratory, The University of Arizona, Tucson, Arizona, United States of America, **8** Pulmonary, Critical Care, and Sleep, Department of Medicine, UC Davis, Sacramento, California, United States of America

* jmosier@aemrc.arizona.edu

**Data Availability Statement:** Data cannot be shared publicly per University of Arizona Health Sciences Research Administration and Banner

## Abstract

### Background

Noninvasive respiratory support modalities are common alternatives to mechanical ventilation in acute hypoxemic respiratory failure. However, studies historically compare noninvasive respiratory support to conventional oxygen rather than mechanical ventilation. In this study, we compared outcomes in patients with acute hypoxemic respiratory failure treated initially with noninvasive respiratory support to patients treated initially with invasive mechanical ventilation.

### Methods

This is a retrospective observational cohort study between January 1, 2018 and December 31, 2019 at a large healthcare network in the United States. We used a validated phenotyping algorithm to classify adult patients (≥18 years) with eligible International Classification of Diseases codes into two cohorts: those treated initially with noninvasive respiratory support or those treated invasive mechanical ventilation only. The primary outcome was time-to-in-hospital death analyzed using an inverse probability of treatment weighted Cox model adjusted for potential confounders. Secondary outcomes included time-to-hospital discharge alive. A secondary analysis was conducted to examine potential differences between noninvasive positive pressure ventilation and nasal high flow.

### Results

During the study period, 3177 patients met inclusion criteria (40% invasive mechanical ventilation, 60% noninvasive respiratory support). Initial noninvasive respiratory support was

Health Research Administration restrictions because of the data contain potentially identifying or sensitive patient information, and are owned by the Banner Health clinical data warehouse. Data are available for researchers who meet the criteria for access to confidential data and with institutional review board approval and a negotiated data use agreement with the University of Arizona. Please direct data use agreement requests to the University of Arizona Health Sciences Research Administration at UAHSContracts@email.arizona.edu.

**Funding:** This work was supported by: E.20124, J.M.M, Emergency Medicine Foundation, Fisher & Paykel Healthcare Directed Grant. Authors supported: Mosier, Subbian, Fisher. https://www.emfoundation.org/grantee/past-grantees/Grantees-2020-2021 1838745, V.S, National Science Foundation. Authors supported: Subbian. https://www.nsf.gov/awardsearch/showAward?AWD_ID=1838745 5T32HL007955, P.E, National Heart, Lung, and Blood Institute, Authors supported: Essay. https://reporter.nih.gov/project-details/8680297 UL1 TR001860, J.C.S, National Center for Advancing Translational Sciences. Authors supported: Stocking. https://reporter.nih.gov/search/pzhJ4Z4AoSUeIr6CoWSajpg/projects K01HL168222, J.C.S, National Heart, Lung, And Blood Institute. Authors supported: Stocking. https://reporter.nih.gov/search/z70wGdIt2ESfrAcjDewhGA/project-details/10643357 The funders had no role in study design, data collection and analysis, decision to publish, or preparation of the manuscript. There was no additional external funding received for this study.

**Competing interests:** Dr. Mosier's competing interest statement has been amended to the following: JMM has received meeting travel support from Fisher & Paykel Healthcare. This does not alter our adherence to PLOS ONE policies on sharing data and materials.

not associated with a decreased hazard of in-hospital death (HR: 0.65, 95% CI: 0.35–1.2), but was associated with an increased hazard of discharge alive (HR: 2.26, 95% CI: 1.92–2.67). In-hospital death varied between the nasal high flow (HR 3.27, 95% CI: 1.43–7.45) and noninvasive positive pressure ventilation (HR 0.52, 95% CI 0.25–1.07), but both were associated with increased likelihood of discharge alive (nasal high flow HR 2.12, 95 CI: 1.25–3.57; noninvasive positive pressure ventilation HR 2.29, 95% CI: 1.92–2.74).

## Conclusions

These data show that noninvasive respiratory support is not associated with reduced hazards of in-hospital death but is associated with hospital discharge alive.

## Introduction

Noninvasive respiratory support strategies utilize an external interface (e.g., facemask, helmet, nasal cannula) to deliver either pressure-based support in the form of continuous or bilevel positive airway pressures; or flow-based support in the form of nasal high flow. Noninvasive modalities, particularly pressure-based support, are recommended for acute exacerbations of chronic obstructive pulmonary disease or acute cardiogenic pulmonary edema [1]. Noninvasive respiratory support modalities are also increasingly used for patients with acute *de novo* hypoxemic respiratory failure, despite unclear data on which strategies are superior and safer, or the impact on outcomes [2–5].

Overall, noninvasive strategies likely reduce the need for intubation and consequently lower mortality compared to standard oxygen [2, 6–8]. However, intubation after failed noninvasive respiratory support is associated with prolonged intensive care unit (ICU) stays and excess mortality [9–16]. The main theory is that nonintubated patients with acute hypoxemic respiratory failure may produce injurious transpulmonary pressures that are inhomogeneously amplified [17] and accelerate lung injury (i.e., patient self-inflicted lung injury). Noninvasive modalities are typically compared to conventional oxygen with the primary outcome of intubation, either alone or in combination with mortality. The benefits of noninvasive respiratory support (improved respiratory mechanics, reduced work of breathing, and improved gas exchange), however, render noninvasive strategies a more appropriate comparison to mechanical ventilation. Data comparing noninvasive respiratory support to invasive mechanical ventilation after conventional oxygen, however, are lacking. The goal of this study was to explore that comparison by investigating the outcomes in patients with acute hypoxemic respiratory failure treated with initial noninvasive respiratory support compared to invasive mechanical ventilation.

## Methods

### Study design, setting, and participants

This retrospective cohort study used de-identified structured clinical data from the Banner Health Network clinical data warehouse. Banner Health spans 26 hospitals across six states in the western United States and uses the Cerner Millennium (Oracle Health, formerly Cerner Corporation, North Kansas City, MO, USA) electronic health record system. Data for all adult patients (≥18 years) admitted to the hospital between January 1, 2018 and December 31, 2019 were extracted on September 16, 2020. Patients were included if they had an admission

diagnosis consistent with the pertinent *International Classification of Diseases* (version 10) sub-codes for acute hypoxemic respiratory failure (J.96): J96.00, J96.01, J96.02, J96.20, J96.21, J96.22, J96.90, J96.91, and J96.92. Patients were excluded if they had a first treatment location other than emergency department, intensive care unit, stepdown unit, or medical/surgical unit. This work adheres to the STROBE reporting guidelines, guidelines from journal editors [18], and was approved by the University of Arizona (#1907780973) and Banner Health Institutional Review Boards (#483-20-0018).

## Cohort assignment

We used a validated phenotyping algorithm to classify eligible cases by the sequence of respiratory support [19–21] into two cohorts: those treated initially with noninvasive respiratory support and those treated initially with invasive mechanical ventilation. All patients in both cohorts were included in the analysis, as there is variation in the determination of failure and physiologic thresholds that prompt intubation [22]. Patients on conventional oxygen only were excluded. Secondary analyses were conducted separating noninvasive respiratory support into noninvasive positive pressure ventilation (either continuous or bilevel positive airway pressure) and nasal high flow. Noninvasive positive pressure ventilation, in any form, in the Banner Health System is provided using a noninvasive ventilator, and nasal high flow is delivered by either the Vapotherm (Vapotherm, Exeter, New Hampshire) system or the OptiFlow (Fisher & Paykel, Auckland, New Zealand) with or without the AirVo™ 2 system.

A subset of patients received noninvasive positive pressure ventilation, nasal high flow, and invasive mechanical ventilation. These patients were manually assigned to the appropriate cohort based on treatment start times and included for analyses comparing noninvasive support to mechanical ventilation but were excluded from analyses separating noninvasive support into noninvasive positive pressure ventilation and nasal high flow.

We estimated the propensity for invasive mechanical ventilation or noninvasive respiratory support (noninvasive positive pressure ventilation or nasal high flow separately in secondary analyses) by using generalized boosted models and used inverse probability of treatment weighting in the models to account for non-random treatment assignment [23], mirroring our previous comparisons in patients with COVID-19 associated respiratory failure [20]. The variables for propensity score estimation included age, body mass index, sex, ethnicity (non-Hispanic, Hispanic), race (white, other), respiratory rate and $SpO_2/FiO_2$ ratio immediately prior to first treatment, comorbidities (diabetes, chronic kidney disease, heart failure, hypertension, chronic obstructive pulmonary disease, neoplasm/immunosuppression, chronic liver disease, obesity), diagnoses of influenza or sepsis, vasopressor infusion before first treatment, first treatment location (emergency department, intensive care unit, stepdown, medical/surgical unit), hospital, time period of hospital admission (time period 1 [January 1—June 30, 2018], time period 2 [July 1—December 31, 2018], time period 3 [January 1—June 30, 2019], and time period 4 [July 1—December 31, 2019]), and hours from hospital admission to first treatment, transformed via the Box-Cox method with negatives [24]. Hospitals with <30 observations were grouped together for ease of modeling and to preserve de-identification. These variables were additionally included in later modeling to further improve balance between treatment groups.

## Outcomes and data analysis

The primary outcome was time-to-in-hospital death, defined as time from initiation of respiratory support to death with hospital discharge considered a competing event. It was modeled using a cause-specific Cox model with the first treatment (noninvasive respiratory support

versus invasive mechanical ventilation) as the key predictor. A secondary outcome of time-to-hospital discharge alive was also evaluated using the same method. Each outcome was also evaluated with secondary analyses separating noninvasive respiratory support into noninvasive positive pressure ventilation and nasal high flow and a sensitivity analysis that removed patients with evidence of all three treatments. We assessed the proportional hazard assumption in the Cox models by including an interaction of time with first treatment and reported the model with the interaction if it was statistically significant at $\alpha = 0.05$. Then, due to limitations of hazard ratios in the presence of competing events, we estimated cumulative incidence curves associated with each treatment using the Cox model estimates [25]. We explored cumulative incidence curves associated with the comorbidities of heart failure and chronic obstructive pulmonary disease and diagnoses of influenza or sepsis both individually and in combination. We set the remaining covariate values to their sample median (continuous covariates) or most frequent value (categorical covariates) [26, 27]. The unweighted outcomes of mortality, intubation rate, days to intubation, and duration of mechanical ventilation were assessed using Fisher's Exact and Kruskal-Wallis rank sum tests where appropriate. We also conducted a subgroup analysis on the primary outcome after excluding patients with a comorbidity of chronic obstructive pulmonary disease or a body mass index >35. Finally, E values for the hazard ratios and hazard ratio confidence limits closest to one are shown in order to estimate how much residual confounding would need to be present to have the true hazard ratio (or limit) be one; the higher the E value, the larger the residual confounding would need to be [28].

Electronic health record data requires accounting for varying levels of missingness among variables [29–31]. Missing data were handled by using multiple imputation by chained equations [32, 33]. For each analysis, we created 50 imputed data sets using all variables in the propensity score, the Nelson-Aalen estimate of the cumulative hazard rate function of available time-to-event data, the time-to-event, and event (i.e., in-hospital death, hospital discharge alive). Body mass index, $SpO_2/FiO_2$, and respiratory rate were imputed via predictive mean matching. Sex, ethnicity, race, and comorbidities were imputed with logistic regression. All variables used for propensity score estimation and the outcome variables were used to model any variables with missing data with the exception that raw time-to-event was not used to predict other variables in the multiple imputation by chained equations algorithm. Instead, temporal information was used for predicting missing values via the Nelson-Aalen estimate. We estimated propensity scores for each imputed data set separately. For the Cox models, the propensity scores from a specific imputed data set were used for inverse probability of treatment weighting for that data set [32, 33], and results were combined using Rubin's Rules. All data preprocessing and statistical analyses were done using R version 4.1.0 [34] with the following packages: twang [35], survival [36, 37], survminer [38], mice [32], xtable [39], and tidyverse [40]. Further detailed descriptions of data preprocessing can be found in our previous work [20].

## Results

There were 3177 patients who met the inclusion criteria. Of these, 1266 (40%) were intubated initially while 1911 (60%) were initially treated with noninvasive respiratory support, Fig 1 and Table 1. There are important differences between cohorts. Patients intubated initially were more commonly male (55% vs. 49%) and disproportionately at large hospitals (57% vs. 44%). They were also of higher acuity based on median APACHE score (68 vs. 49), although only patients admitted to an ICU were given an APACHE score in the electronic health record. Those intubated initially were less likely to have comorbid heart failure (36% vs. 47%) or

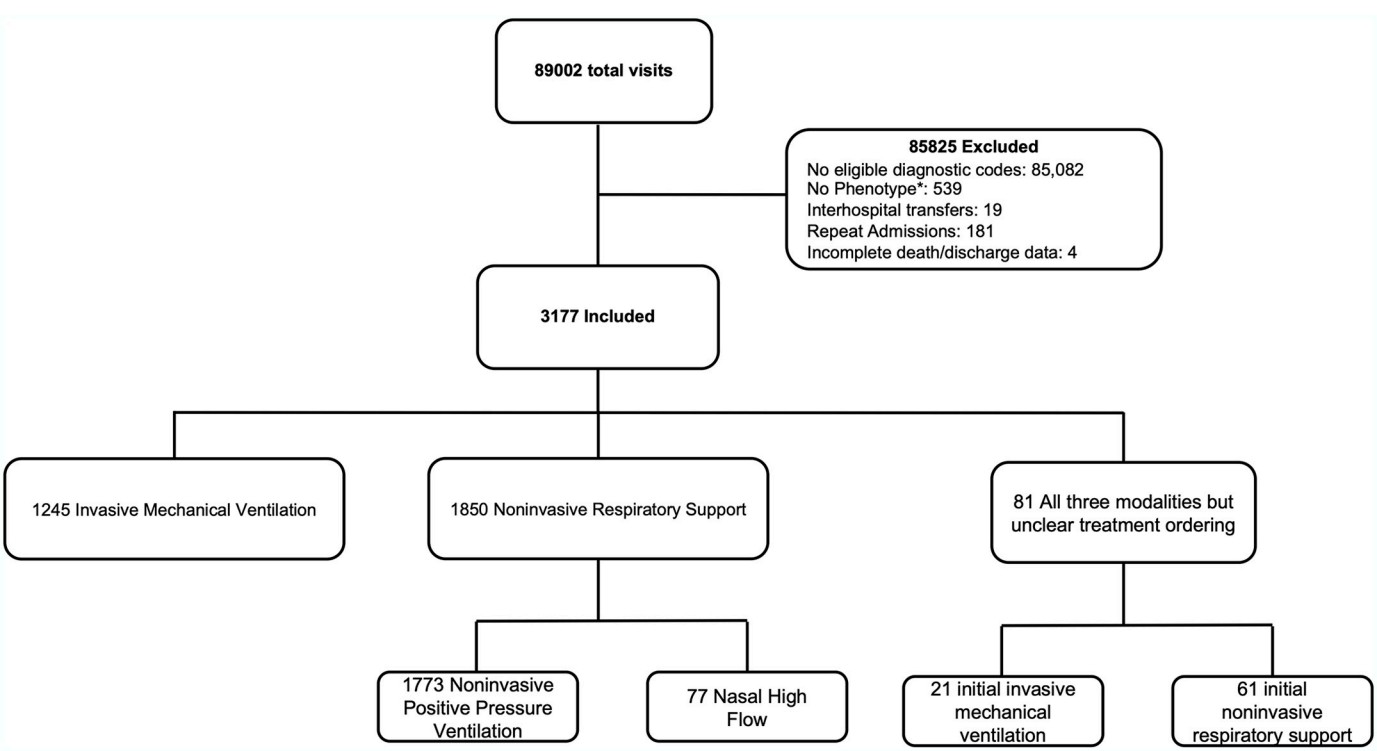

**Fig 1. STROBE diagram of included subjects.** There were 89,002 total visits during the study period. Of those, most (85,825) failed to meet exclusion criteria. *The subjects that were excluded because they were not classified by the algorithm but had an eligible diagnostic code on admission likely represent those only requiring conventional oxygen. Repeat admissions and interhospital transfers were excluded due to confounding with the outcome.

chronic obstructive pulmonary disease (68% vs. 85%), more likely to be septic (36% vs. 19%), and had more severe hypoxemia based on $SpO_2/FiO_2$ on treatment assignment (medians 130 vs. 261, difference of means 90.52, 95% CI: 84.04–97.01) despite clinically similar worst $PaO_2/FiO_2$ in the first 24 hours (medians 124 vs. 142, difference of means 4.59, 95% CI: -4.39–13.58). Of the 1850 (97%) patients where the sequence of noninvasive respiratory support could be reliably classified, most patients (96%) were treated with noninvasive positive pressure ventilation (S1 Table: **Demographics**). All-cause in-hospital mortality was 13%, higher for patients intubated first than for those treated with noninvasive respiratory support first (18% vs 9%), S2 Table: **Unmatched Outcomes**. Mortality was significantly higher for noninvasive respiratory support patients who required intubation compared to those who did not (22% vs. 6%).

Hazard ratios are shown in Table 2. Initial noninvasive respiratory support was not associated with a decreased hazard of in-hospital death (HR: 0.65, 95% CI: 0.35–1.2); with no significant interaction of treatment and time and a reasonable protection from unmeasured confounders with an E-value of 2.04. Initial noninvasive respiratory support was, however, associated with an increased hazard of discharge alive (HR: 2.26, 95% CI: 1.92–2.67) that decreased over time (interaction between time and noninvasive respiratory support HR: 0.97, 95% CI: 0.95–0.98) and an even stronger E-value (2.90), Fig 2. The subgroup analysis excluding patients with comorbid chronic obstructive pulmonary disease or a body mass index >35 showed no statistically significant association between initial respiratory support modality and time to in-hospital death, but the hazard ratio increased for noninvasive respiratory support compared to mechanical ventilation (1.71, 95% CI: 0.85–3.45), with an E-value of 2.25. Sensitivity analyses excluding the patients without clear treatment sequence shows noninvasive

**Table 1. Demographics.**

| Measure | Invasive Mechanical Ventilation | Noninvasive Respiratory Support | Total |
|---|---|---|---|
| N (%) | 1266 (40%) | 1911 (60%) | 3177 |
| Female Sex | 574 (45%) | 970 (51%) | 1544 (49%) |
| Age, median (IQR) | 61 (48–72) | 68 (58–77) | 66 (54–75) |
| BMI, median (IQR) | 28 (23–35) | 29 (23–37) | 28 (23–36) |
| Ethnicity, n(%)[a] | | | |
| Not Hispanic or Latino | 1031 (82%) | 1645 (86%) | 2676 (85%) |
| Hispanic or Latino | 225 (18%) | 261 (14%) | 486 (15%) |
| Race, n (%)[a] | | | |
| White | 1078 (82%) | 1690 (89%) | 2768 (88%) |
| Black or African American | 76 (6%) | 105 (6%) | 181 (6%) |
| Asian/Native Hawaiian/Pacific Islander | 16 (1%) | 18 (1%) | 34 (1%) |
| American Indian or Alaska Native | 62 (5%) | 39 (2%) | 101 (3%) |
| Other | 25 (2%) | 53 (3%) | 78 (2%) |
| Hospital Size, n(%) | | | |
| small | 39 (4%) | 158 (9%) | 197 (7%) |
| medium | 428 (39%) | 815 (47%) | 1243 (44%) |
| large | 631 (57%) | 779 (44%) | 1410 (49%) |
| APACHE IVa median (IQR) | 68 (50–87) | 49 (38–63) | 57 (43–76) |
| Vital Signs on Treatment Assignment median (IQR) | | | |
| Heart rate | 93 (79–112) | 88 (75–104) | 90 (76–107) |
| Systolic blood pressure | 120 (103–140) | 132 (116–149) | 127 (111–146) |
| Diastolic blood pressure | 68 (57–82) | 73 (64–83) | 71 (61–83) |
| SpO2 | 98 (96–100) | 96 (93–98) | 97 (94–99) |
| FiO2[b] | 75 (50–100) | 36 (30–50) | 45 (33–80) |
| SpO2:FiO2 | 130 (99–200) | 261 (186–323) | 200 (116–297) |
| Temperature (˚C) | 37 (36.6–37) | 36.8 (36.5–37) | 36.9 (36.5–37) |
| Respiratory Rate | 19 (16–23) | 20 (18–25) | 20 (18–24) |
| Comorbidities n(%) | | | |
| Diabetes | 468 (40%) | 756 (40%) | 1224 (40%) |
| Chronic Kidney Disease | 261 (22%) | 527 (28%) | 788 (26%) |
| Heart Failure | 418 (36%) | 903 (47%) | 1321 (43%) |
| Hypertension | 839 (72%) | 1427 (75%) | 2266 (74%) |
| Chronic Liver Disease | 242 (21%) | 157 (8%) | 399 (13%) |
| Neoplasm or Immunosuppression | 175 (15%) | 293 (15%) | 468 (15%) |
| COPD | 799 (68%) | 1613 (85%) | 2412 (78%) |
| Obesity | 186 (16%) | 350 (18%) | 536 (17%) |
| Acute Influenza Diagnosis | 2 (0%) | 4 (0%) | 6 (0%) |
| Acute sepsis diagnosis | 419 (36%) | 354 (19%) | 773 (25%) |
| Labs on Admission median (IQR) | | | |
| $PaO_2$ (mmHg) (Worst Value) | 79 (65–107) | 75 (63–102) | 77 (63–104) |
| $PaO_2$:$FiO_2$ (Worst Value) | 124 (79–220) | 142 (84–218) | 133 (81–218) |
| White Blood Cell Count (K/uL) | 8 (2.45–14) | 7.7 (2–13) | 7.9 (2–13.5) |
| Lactate (mmol/L) | 1.9 (1.2–3.7) | 1.5 (1–2.3) | 1.7 (1.1–2.8) |
| pH | 7.31 (7.19–7.39) | 7.33 (7.26–7.41) | 7.33 (7.24–7.4) |
| $PaCO_2$ (mmHg) | 46 (38–64) | 53 (40–70) | 50 (38–68) |
| $HCO_3$ (mmol/L) | 24 (20–28) | 27 (23–32) | 26 (22–30) |
| BNP (pg/mL) | 1490 (359–5781) | 1316 (327–5381) | 1369 (334–5476) |

*(Continued)*

**Table 1.** (Continued)

| Measure | Invasive Mechanical Ventilation | Noninvasive Respiratory Support | Total |
|---|---|---|---|
| Creatinine (mg/dL) | 1.0 (0.76–1.58) | 0.95 (0.73–1.36) | 0.97 (0.74–1.43) |
| Time from hospital admission to treatment (h) | 0.6 (0–7) | 27 (0–89) | 6 (0–61) |
| Treatment Assignment Location n (%)[a] | | | |
| Emergency Department | 666 (53%) | 523 (27%) | 1189 (37%) |
| ICU | 505 (40%) | 644 (34%) | 1149 (36%) |
| Non-ICU ward | 16 (1%) | 272 (14%) | 288 (9%) |
| Stepdown | 79 (6%) | 472 (25%) | 551 (17%) |

[a] Data are presented as percent of available.

[b] $FiO_2$ determined by documented $FiO_2$, if documented, or by $FiO_2 = 100(0.21 + $ oxygen flow $[L/min^{-1}])$ x 0.03 if a flow rate was documented.

respiratory support was associated with a reduced hazard of in-hospital death (HR: 0.53, 95% CI: 0.28–0.99, E-value 2.49) and an increased hazard of hospital discharge alive (HR: 2.34, 95% CI: 1.97–2.77, E-value 2.98) that again decreased over time (interaction between time and noninvasive respiratory support HR: 0.97, 95% CI: 0.95–0.98). Representative cumulative incidence curves show a consistent trend of slightly reduced probability of in-hospital death and increased probability of discharge alive for noninvasive respiratory support across a range of comorbidities (chronic obstructive pulmonary disease, congestive heart failure) and diagnoses (influenza, sepsis), except for the subgroup analysis S1–S5 Figs. The cumulative incidence

**Table 2. Cox model results.**

| Outcome | | First Treatment | Hazard Ratio | 95% CI | P-Value | E-Value | CL E-Value |
|---|---|---|---|---|---|---|---|
| **Time to In-Hospital Death** | Primary analysis | NIRS | 0.65 | 0.35–1.2 | 0.167 | 2.04 | 1.00 |
| | Sensitivity analysis | NIRS | 0.53 | 0.28, 0.99 | 0.047 | 2.49 | 1.08 |
| | Subgroup analysis* | NIRS | 1.71 | 0.85, 3.45 | 0.136 | 2.25 | 1.00 |
| | Secondary analysis | NHF | 3.27 | 1.43, 7.45 | 0.005 | 3.91 | 1.89 |
| | | NIPPV | 0.52 | 0.25, 1.07 | 0.076 | 2.53 | 1.00 |
| **Time to Hospital Discharge Alive** | Primary analysis | NIRS | 2.26 | 1.92, 2.67 | <0.001 | 2.90 | 2.51 |
| | | Time x NIRS | 0.97 | 0.95, 0.98 | <0.001 | 1.18 | 1.13 |
| | Sensitivity analysis | NIRS | 2.34 | 1.97, 2.77 | <0.001 | 2.98 | 2.57 |
| | | Time x NIRS | 0.97 | 0.95, 0.98 | <0.001 | 1.18 | 1.14 |
| | Secondary analysis | NHF | 2.12 | 1.25, 3.57 | 0.005 | 2.74 | 1.62 |
| | | NIPPV | 2.29 | 1.92, 2.74 | <0.001 | 2.94 | 2.51 |
| | | Time x NHF | 0.99 | 0.93, 1.05 | 0.795 | 1.08 | 1.00 |
| | | Time x NIPPV | 0.96 | 0.95, 0.98 | <0.001 | 1.19 | 1.14 |

NIRS = Noninvasive respiratory support

NHF = nasal high flow

NIPPV = noninvasive positive pressure ventilation

* excluding patients with comorbid chronic obstructive pulmonary disease or a body mass index >35. Note that for this analysis, we had to remove influenza from the propensity score calculation and regroup some hospitals due to the reduced sample size.

Additional predictors include age, BMI, gender, ethnicity, white race, respiratory rate (breaths/min), the ratio of $SPO_2$ to $FIO_2$, diabetes, CKD, hypertension, heart failure, COPD, neoplasm or immunosuppression, chronic liver disease, obesity, influenza, sepsis, vasopressors before treatment, transformed first treatment start (in days after hospital admission), first treatment location type (levels ED, ICU, Stepdown, or Med/Surg), hospital (with a grouped level for hospitals with very few patients in the data set), and time period.

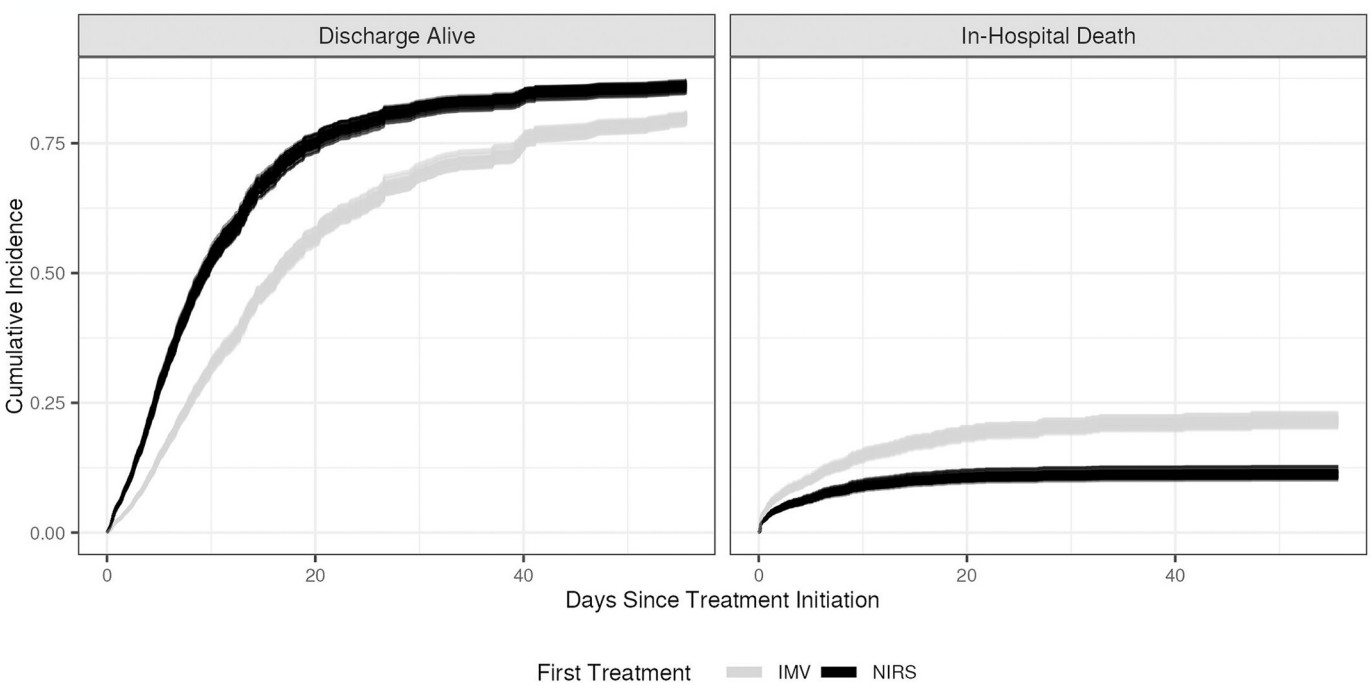

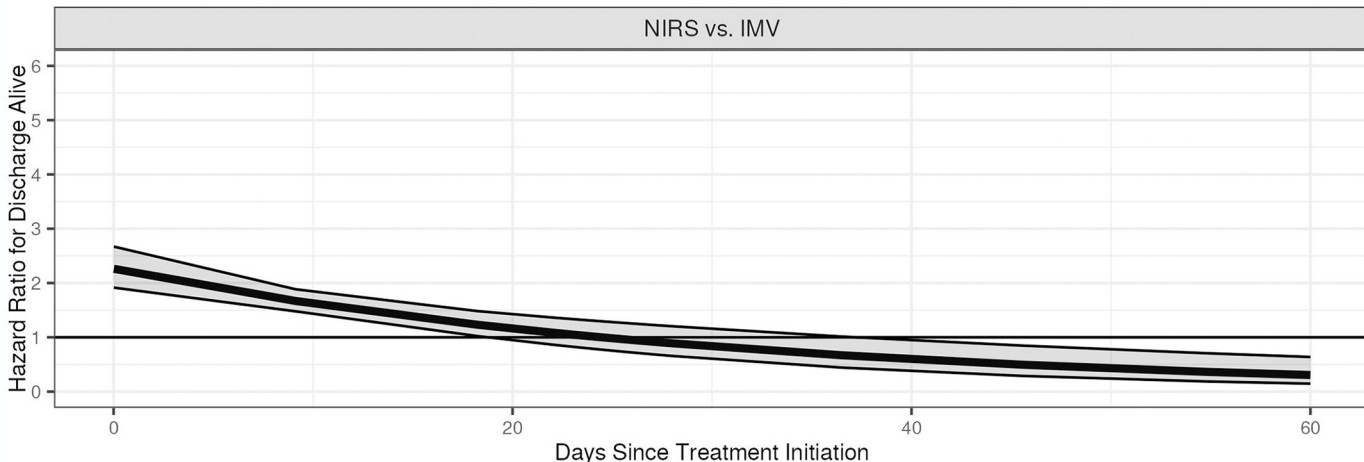

**Fig 2.** Top: Model-estimated cumulative incidence curves for noninvasive respiratory support (NIRS) vs invasive mechanical ventilation (IMV) showing the probabilities for hospital discharge alive (left) and in-hospital death (right). Bottom: Estimated time-varying hospital discharge alive hazard ratios for NIRS versus IMV with pointwise 95% confidence intervals. The following values were used for covariates: male, not Hispanic or Latino, white, one of the large hospitals (hospital A), hospital admission to the emergency department between January 1, 2018 and June 30, 2018, no vasopressor infusion before treatment, no diabetes, no chronic kidney disease, no heart failure, yes hypertension, no chronic obstructive pulmonary disease, no neoplasm/immunosuppression, no chronic liver disease, no obesity, no influenza, yes sepsis, and continuous covariates set at their median values (age = 66 years, $SpO_2/FiO_2$ = 200, respiratory rate = 20 breaths/min, BMI = 28.44, transformed hours from hospital admission to first treatment = 1.77). Each imputed data set generates a pair of curves (one for IMV, one for NIRS).

curves in the subgroup analysis suggest an increased probability of in-hospital death with non-invasive respiratory support.

The probability of discharge alive is greater for NIRS than for IMV, and the probability of in-hospital death is lower. The hazard ratio of discharge alive for NIRS vs. IMV starts out statistically significantly positive shortly before 20 days after treatment initiation and eventually switches direction around 40 days.

In the secondary analyses separating noninvasive respiratory support modalities, the hazard for in-hospital death ranged from increased with nasal high flow (HR 3.27, 95% CI: 1.43–7.45, E-value 3.91) to non-significantly decreased with noninvasive positive pressure ventilation (HR 0.52, 95% CI 0.25–1.07, E-value 2.53), neither with an interaction with time, Fig 3. Both modalities were associated with increased hazard of discharge alive (nasal high flow HR 2.12, 95 CI: 1.25–3.57, E-value 2.74; noninvasive positive pressure ventilation HR 2.29, 95% CI: 1.92–2.74, E-value 2.94), both with strong E-values, but only noninvasive positive pressure ventilation showed an interaction with time (interaction between time and nasal high flow HR 0.99, 95% CI: 0.93–1.05; interaction between time and noninvasive positive pressure

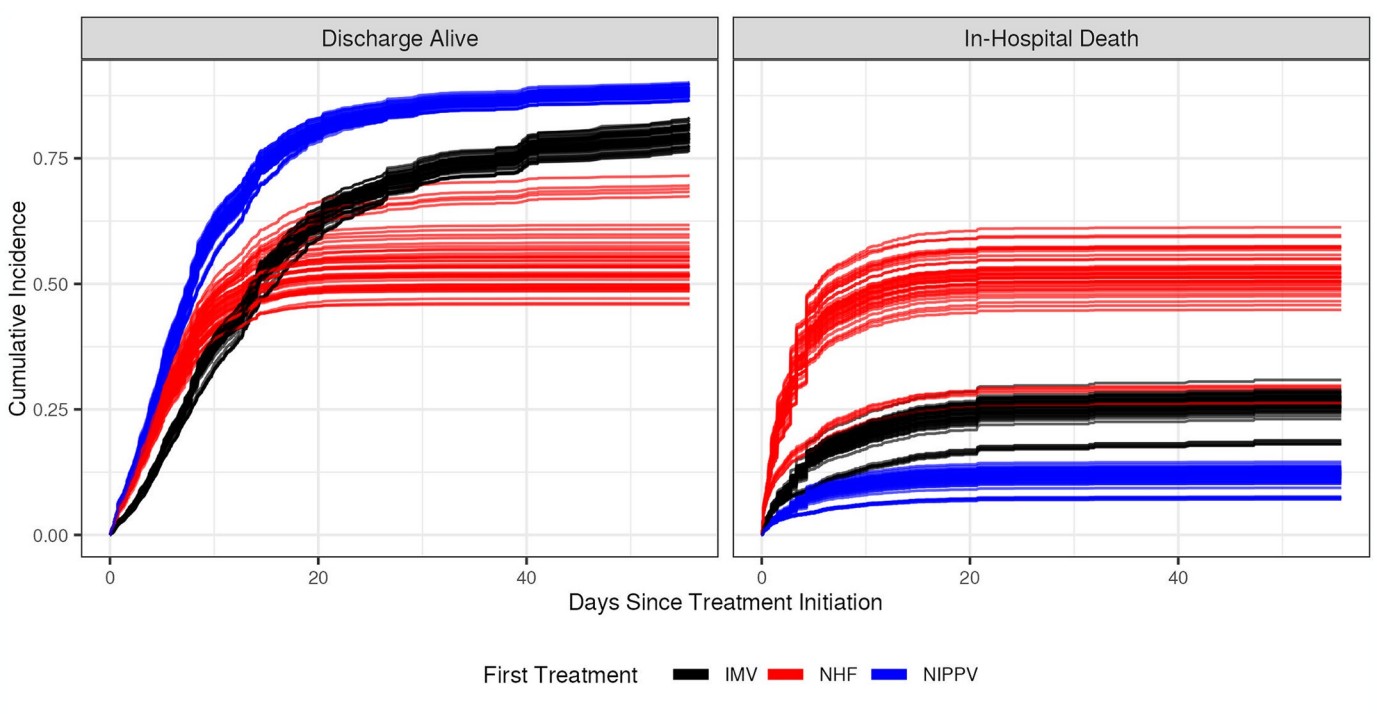

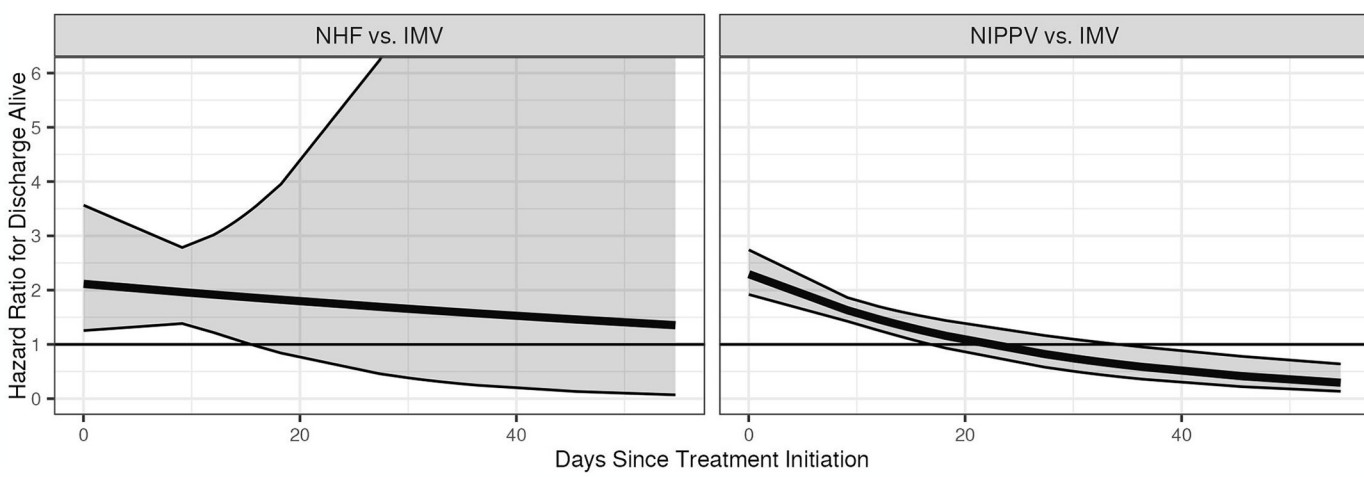

**Fig 3.** Top: Model-estimated cumulative incidence curves for noninvasive positive pressure ventilation (NIPPV), nasal high flow (NHF), and invasive mechanical ventilation (IMV) showing the probabilities for hospital discharge alive (left) and in-hospital death (right). Bottom: Estimated time-varying hospital discharge alive hazard ratios for NHF versus IMV (left) and NIPPV versus IMV (right) with pointwise 95% confidence intervals.

ventilation HR 0.96, 95% CI: 0.95–0.98). Representative cumulative incidence curves are shown in S6 and S7 Figs.

The following values were used for covariates: male, not Hispanic or Latino, white, one of the large hospitals (hospital A), hospital admission to the emergency department between January 1, 2018 and June 30, 2018, no vasopressor infusion before treatment, no diabetes, no chronic kidney disease, no heart failure, yes hypertension, no chronic obstructive pulmonary disease, no neoplasm/immunosuppression, no chronic liver disease, no obesity, no influenza, yes sepsis, and continuous covariates set at their median values (age = 66 years, $SpO_2/FiO_2$ = 200, respiratory rate = 20 breaths/min, BMI = 28.44, transformed hours from hospital admission to first treatment = 1.70). Each imputed data set generates a triple of curves (one for IMV, NHF, and NIPPV).

Patients initially treated with either non-invasive modality had a higher probability of hospital discharge alive until roughly 15 days after treatment initiation. After that, the probability of discharge alive remained higher for NIPPV compared to IMV but reversed direction for NHF and IMV. Nasal high flow had a slightly higher probability of in-hospital death than IMV, and NIPPV had a slightly lower probability of in-hospital death than IMV. The hospital discharge alive hazard ratio of NHF to IMV was positive but decreasing until around day 15, at which point there was no further clear difference between those two treatments. The same pattern held for the hospital discharge alive hazard ratio of NIPPV to IMV except that it eventually reversed direction, resulting in the hazard of hospital discharge alive being greater for IMV than NIPPV starting at roughly 35 days after treatment initiation.

## Discussion

Noninvasive respiratory support strategies are increasingly used as alternative initial strategies to early intubation and mechanical ventilation for patients with acute hypoxemic respiratory failure. Thus, the goal of this study was to compare the outcomes between those two approaches. Our results show that initial noninvasive respiratory support modalities in patients with acute hypoxemic respiratory failure were most likely not associated with a reduced hazard of in-hospital death (no association in the primary analyses, weakly significant association in the sensitivity analysis), yet were associated with an increased probability of hospital discharge alive when compared to initial invasive mechanical ventilation. The existing literature comparing noninvasive strategies to conventional oxygen suggests that noninvasive respiratory support is probably associated with reduced mortality, reduced intubation, and shorter hospitals stays, but to varying degrees among the different noninvasive modalities [2, 4, 5, 41, 42]. Our results expand upon this knowledge by comparing outcomes between noninvasive strategies and invasive mechanical ventilation in non-COVID-19 acute hypoxemic respiratory failure. Despite patients who were intubated first generally being more severely ill and more likely to be septic, there was no increase in hazard of our primary outcome (in-hospital death). However, noninvasive respiratory support was associated with an increased likelihood of hospital discharge alive, which waned over time for nasal high flow. Further study may show that failure of a noninvasive respiratory support modality may be associated with increased mortality beyond the progression of disease, thus having an outsized influence on the overall association with mortality [43]. This hypothesis is supported by the observed 3.5-fold increase in unweighted mortality for patients that failed a noninvasive strategy compared to those that did not, and is also suggested by the results of our subgroup analysis.

These data suggest that noninvasive respiratory support modalities can be effective alternatives to mechanical ventilation for the initial treatment of some patients with acute hypoxemic respiratory failure. Successful noninvasive support increases the likelihood of earlier hospital

discharge, but an unsuccessful trial may carry outsized consequences for mortality. These results add to findings from the Lung Safe study, that showed noninvasive positive pressure ventilation was used in 15% of patients with acute respiratory distress syndrome, but both failure and mortality increased as the severity of disease worsened [15]. While spontaneous breathing can have some advantages in acute hypoxemic respiratory failure patients, patient self-inflicted lung injury is the likely reason for worse outcomes in patients that fail a noninvasive strategy [44, 45]. Taking these data and existing literature together, balancing the double-edged sword with noninvasive respiratory support involves early application of noninvasive respiratory support with close monitoring of the patient's work of breathing and avoiding delayed intubation in patients where noninvasive modalities fail to sufficiently reduce the work of breathing [46].

Our secondary analyses also suggested differences between noninvasive positive pressure ventilation and nasal high flow. There are four possible explanations for these findings. The first is that noninvasive positive pressure may be the better noninvasive strategy. The second possibility is that the patients treated with nasal high flow may not have been similar to the patients treated with noninvasive positive pressure ventilation, and that our efforts to account for treatment confounding were not fully successful. Patients treated with nasal high flow in our dataset had a higher median APACHE score on admission (56 vs. 48, difference of means 11.77, 95% CI: 6.36, 17.18) a lower median $SpO_2/FiO_2$ on treatment assignment (136 vs 274, difference of means -105, 95% CI: -126 - -85) and lower median worst $PaO_2/FiO_2$ (76 vs 150, difference of means -76, 95% CI: -106 - -45), more commonly had neoplasm or immunosuppression (26% vs 15%) and were more commonly septic (34% vs 17%). The third possibility is that patients may not have been treated similarly. Patients initiated on noninvasive positive pressure in a non-intensive care unit were more commonly transferred to the intensive care unit by 12 hours than patients started on nasal high flow (32% vs 12%). Lastly, there may have been imbalanced, imprecise, or incorrect delivery of one modality compared to the other. The median flow rate for nasal high flow was 40lpm (95% CI: 35 - 50lpm). While gas exchange can improve at lower flow rates, higher flow rates are required for the work of breathing benefits related to changes in resting lung volume and strain [47]. Monitoring likely differed between intensive care (e.g., work of breathing changes, signs of fatigue) and non-intensive care units (e.g., oxygen saturation). Additionally, managing failure could have differed as failing nasal high flow could have resulted in more crossover to noninvasive positive pressure ventilation than intubation compared to failing noninvasive positive pressure ventilation.

There are important limitations to our results. Our data were limited to pre-COVID-19 data. As such, the use of noninvasive respiratory support modalities has likely evolved as is evident by the relatively low number of patients treated with nasal high flow across the entire health network. Second, we used admission diagnostic codes to select patients treated at the time for acute hypoxemic respiratory failure without the selection bias of using discharge diagnostic codes. These results are contingent upon accurate coding of admission diagnoses and important patient groups may have been excluded. Since these data are non-randomized observational data, non-protocolized clinical care may have contributed unmeasured confounding differences in the selection for and management of each modality. We attempted to control for confounding by inverse probability for treatment assignment weighting and further adjusting for potential confounders in the Cox models. Furthermore, our E-values are relatively strong, so any unmeasured confounding would have needed to have a substantial effect to alter the results. Another limitation is that results are based on the first assigned therapy, and symptom onset time is not available in our dataset. Thus, crossover (and imbalanced crossover), and symptom duration could confound the findings. Lastly, goals of care and end-of-life issues are not readily extractable from structured electronic health record data. There is

an important difference between a patient who is a do-not-intubate on admission treated with rescue noninvasive respiratory support and a similar patient who worsened during noninvasive respiratory support and chose to become a do-not-intubate. However, both patients would have been included in our dataset and could contribute some confounding in the results.

Despite these limitations, our results provide an overview of outcomes between respiratory support modalities that were pragmatically applied across a large healthcare network. These results highlight important knowledge gaps needing further study, including: 1. the risks of failing noninvasive respiratory support, mechanisms of those risks, thresholds of, monitoring for, and management of failure, 2. reproducible phenotypes likely to do well or not do well with noninvasive respiratory support modality, 3. optimal noninvasive support modality by phenotype, 4. optimal noninvasive respiratory support delivery by modality and, 5. optimal hospital location and minimal monitoring capabilities for patients with acute hypoxemic respiratory failure requiring noninvasive respiratory support.

Our data across a large and diverse healthcare network show that initial treatment with noninvasive respiratory support is not associated with a reduced hazard of in-hospital death compared to invasive mechanical ventilation for patients admitted with acute hypoxemic respiratory failure. However, noninvasive respiratory support is associated with a higher likelihood of earlier hospital discharge. Lastly, our data suggest potential differences between noninvasive respiratory support modalities that require further exploration.

## Supporting information

**S1 Table. Demographics of patients without clear sequence of support.**
(DOCX)

**S2 Table. Unmatched outcomes.**
(DOCX)

**S1 Fig. Representative in-hospital death-model cumulative incidence curves.**
(DOCX)

**S2 Fig. Representative in-hospital death-model cumulative incidence curves excluding patients without clear sequence of noninvasive respiratory support.**
(DOCX)

**S3 Fig. Representative in-hospital death-model cumulative incidence curves of a subgroup excluding patients with a comorbidity of COPD or body mass index >35.**
(DOCX)

**S4 Fig. Representative hospital discharge alive model-estimated cumulative incidence curves.**
(DOCX)

**S5 Fig. Representative hospital discharge alive model-estimated cumulative incidence curves excluding patients without clear sequence representative hospital discharge alive model-estimated cumulative incidence curves excluding patients without clear sequence of noninvasive respiratory support.**
(DOCX)

**S6 Fig. Representative in-hospital death model-estimated cumulative incidence curves excluding patients without clear sequence of noninvasive respiratory support.**
(DOCX)

**S7 Fig. Representative hospital discharge alive model-estimated cumulative incidence curves excluding patients without clear sequence of noninvasive respiratory support.** (DOCX)

## Acknowledgments

The authors would like to thank Don Saner and Mario Arteaga from the Banner Health Network Clinical Data Warehouse for their support during this project.

## Author Contributions

**Conceptualization:** Jarrod M. Mosier, Vignesh Subbian, Patrick Essay, Edward J. Bedrick, Jacqueline C. Stocking, Julia M. Fisher.

**Data curation:** Jarrod M. Mosier, Vignesh Subbian, Sarah Pungitore, Devashri Prabhudesai, Patrick Essay, Julia M. Fisher.

**Formal analysis:** Jarrod M. Mosier, Vignesh Subbian, Sarah Pungitore, Devashri Prabhudesai, Patrick Essay, Edward J. Bedrick, Jacqueline C. Stocking, Julia M. Fisher.

**Funding acquisition:** Jarrod M. Mosier, Vignesh Subbian, Patrick Essay, Julia M. Fisher.

**Investigation:** Jarrod M. Mosier, Vignesh Subbian, Sarah Pungitore, Patrick Essay, Julia M. Fisher.

**Methodology:** Jarrod M. Mosier, Vignesh Subbian, Patrick Essay, Edward J. Bedrick, Julia M. Fisher.

**Project administration:** Jarrod M. Mosier.

**Resources:** Vignesh Subbian.

**Supervision:** Jarrod M. Mosier, Vignesh Subbian, Edward J. Bedrick, Julia M. Fisher.

**Validation:** Julia M. Fisher.

**Visualization:** Jarrod M. Mosier, Devashri Prabhudesai, Edward J. Bedrick, Jacqueline C. Stocking, Julia M. Fisher.

**Writing – original draft:** Jarrod M. Mosier, Julia M. Fisher.

**Writing – review & editing:** Jarrod M. Mosier, Vignesh Subbian, Sarah Pungitore, Devashri Prabhudesai, Patrick Essay, Edward J. Bedrick, Jacqueline C. Stocking, Julia M. Fisher.

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
