## [Decision Letter · Decision Letter 0]

24 Mar 2024

PONE-D-24-02817Noninvasive vs invasive respiratory support for patients with acute hypoxemic respiratory failurePLOS ONE

Dear Dr. Mosier,

Thank you for submitting your manuscript to PLOS ONE. After careful consideration, we feel that it has merit but does not fully meet PLOS ONE’s publication criteria as it currently stands. Therefore, we invite you to submit a revised version of the manuscript that addresses the points raised during the review process.

We look forward to receiving your revised manuscript.

Kind regards,

Stefan Grosek, Ph.D., M.D.,

Academic Editor

PLOS ONE

Journal Requirements:

This work was supported by an Emergency Medicine Foundation grant sponsored by Fisher & Paykel, and in part by the National Science Foundation under grant #1838745 and the National Heart, Lung, and Blood Institute of the National Institutes of Health under award number 5T32HL007955. Neither funding agency or sponsor was involved in the design or conduct of the study or interpretation and presentation of the results. 

JMM has received travel support from Fisher & Paykel.

We note that you received funding from a commercial source: Fisher & Paykel.

Additional Editor Comments:

Dear Authors

This retrospective study on patients with acute hypoxemic respiratroy support who were treated as first with invasive or noninvasive ventilation answered to may interesting questions It is well designed and written and brings some interesting answers to yours questions. Both reviewers are of the same opinion as meed but found somme issues to be discussed or elaborated.

I'm looking forward hearing from you soon.

Kind regards.

Reviewers' comments:

Reviewer's Responses to Questions

**Comments to the Author**

1. Is the manuscript technically sound, and do the data support the conclusions?

Reviewer #1: Partly

Reviewer #2: Yes

2. Has the statistical analysis been performed appropriately and rigorously? 

Reviewer #1: Yes

Reviewer #2: Yes

3. Have the authors made all data underlying the findings in their manuscript fully available?

Reviewer #1: Yes

Reviewer #2: Yes

4. Is the manuscript presented in an intelligible fashion and written in standard English?

Reviewer #1: Yes

Reviewer #2: Yes

5. Review Comments to the Author

Reviewer #1: Mosier and co-authors explored the outcomes of patients affected by acute hypoxemic respiratory failure treated with NIRS (noninvasive respiratory supports) compared to patients initially treated with IMV (invasive mechanical ventilation). They found that initial treatment with NIRS is not associated with a reduced in-hospital death when compared to IMV, but NIRS treatment is associated with a higher likelihood of earlier hospital discharge.

The authors should be commended for the work they have performed. However, I believe that there is a major issue to resolve:

It is not clear if cases analyzed are “de novo acute hypoxemic respiratory failure”, exacerbations or a mix of cases. In the introduction section authors correctly referred to the use of NIRS in acute exacerbation of COPD or acute cardiogenic pulmonary oedema, and in “de novo” hypoxemic respiratory failure (lines 94-99, page 3). However, having a look at data presented in table 1, it comes out that the median pH of the included population is 7.31 (IMV) and 7.33 (NIV). This potentially correlates with the level of CO2 found (46 vs 53). Taking into account that bicarbonate levels were pretty normal or even higher in the noninvasive group, this means that most of the patients were affected already by a chronic respiratory condition. Indeed, 78% and 16% in IMV and 85% and 18% in noninvasive group were respectively COPD or obese. Surprisingly, only 0% was affected by acute respiratory conditions as influenza, and no ARDS patients were enlisted in the retrieved data.

In conclusion, it seems that there is no distinction between de novo and acute exacerbations of chronic condition, and that the majority of this cohort of patients analyzed is affected by exacerbation of chronic condition. I believe that this distinction and a subanalysis of specific populations should also be considered (obesity, COPD, etc).

Reviewer #2: This is an interesting retrospective comparison of in-hospital mortality and the discharge alive of patients with acute respiratory failure treated initially either with noninvasive respiratory support or with invasive mechanical ventilation and deserves to be published. Noninvasive respiratory was associated with more discharges alive, but no reduction in in-hospital mortality at the expense of the high-flow nasal oxygen (HFNO) therapy group. Due to the retrospective nature of the study, treatment with NHFO was problematic and inadequate in relation to the need for respiratory support. Thus, the results do not reflect the actual effectiveness of NHO therapy. It is true, however, that the data comes from real life, even from the pre-Covid-19 period. In addition to the benefits of HFNO in preventing intubation, prior studies have already demonstated the deleterous effect with inadequate use of HFNO. Meanwhile, the knowledge and the way of using NHFO has improved. The authors have already discussed in details the limitations of the study. One of the important issues is the imbalance of the groups. HFNO was the ceiling treatment for terminal-stage diseased patients irrespective of the severity of respiratory failure. Thus, the inclusion criteria for NIPPV group and HFNO group differ significantly as does the outcome.

The authors should address the perspectives of the study in discussion: When is it safe to start HFNO in acute respiratory failure? When to escalate from HFNO to NIPPV or to MV? Is a stepwise approach the right one?

6. PLOS authors have the option to publish the peer review history of their article (what does this mean?). If published, this will include your full peer review and any attached files.

Reviewer #1: No

Reviewer #2: No

---

## [Author Response · Author response to Decision Letter 0]

27 Apr 2024

PONE-D-24-02817

Noninvasive vs invasive respiratory support for patients with acute hypoxemic respiratory failure

PLOS ONE

Dear Dr. Grosek and editorial board,

Thank you for the opportunity to revise our manuscript. We thank the reviewers for the excellent comments and think they have strengthened our paper significantly. Below is a point-by-point response for your consideration. We hope you find these revisions to be acceptable for publication in PLOS ONE and we look forward to your decisions. 

On behalf of all authors, 

Jarrod Mosier, MD

Professor and Vice Chair of Emergency Medicine 

Professor of Medicine

Department of Emergency Medicine

Department of Medicine, Division of Pulmonary, Allergy, Critical Care and Sleep

University of Arizona College of Medicine-Tucson 

Journal Requirements:

https://journals.plos.org/plosone/s/file?id=wjVg/PLOSOne_formatting_sample_main_body.pdfand

Response: The manuscript has been edited to meet the additional formatting requirements. 

This work was supported by an Emergency Medicine Foundation grant sponsored by Fisher & Paykel, and in part by the National Science Foundation under grant #1838745 and the National Heart, Lung, and Blood Institute of the National Institutes of Health under award number 5T32HL007955. Neither funding agency or sponsor was involved in the design or conduct of the study or interpretation and presentation of the results. 

Response: Our funding statement has been updated in the cover letter to the following: 

This work was supported by: 

E.20124, J.M.M, Emergency Medicine Foundation, Fisher & Paykel Healthcare Directed Grant. Authors supported: Mosier, Subbian, Fisher. https://www.emfoundation.org/grantee/past-grantees/Grantees-2020-2021

1838745, V.S, National Science Foundation. Authors supported: Subbian. https://www.nsf.gov/awardsearch/showAward?AWD_ID=1838745

5T32HL007955, P.E, National Heart, Lung, and Blood Institute, Authors supported: Essay. https://reporter.nih.gov/project-details/8680297

UL1 TR001860, J.C.S, National Center for Advancing Translational Sciences. Authors supported: Stocking. https://reporter.nih.gov/search/pzhJ4ZAoSUelr6CoWSajpg/projects

K01HL168222, J.C.S, National Heart, Lung, And Blood Institute. Authors supported: Stocking. https://reporter.nih.gov/search/z70wGdlt2ESfrAcjDewhGA/project-details/10643357

There was no additional external funding received for this study.

JMM has received travel support from Fisher & Paykel.

We note that you received funding from a commercial source: Fisher & Paykel.

Response: Dr. Mosier’s competing interest statement has been amended to the following: 

JMM has received meeting travel support from Fisher & Paykel Healthcare. This does not alter our adherence to PLOS ONE policies on sharing data and materials. 

Additional Editor Comments:

Dear Authors

This retrospective study on patients with acute hypoxemic respiratroy support who were treated as first with invasive or noninvasive ventilation answered to may interesting questions It is well designed and written and brings some interesting answers to yours questions. Both reviewers are of the same opinion as meed but found somme issues to be discussed or elaborated.

I'm looking forward hearing from you soon.

Kind regards.

Response: Thank you for the kind feedback. We hope you find the revision acceptable. 

Reviewers' comments:

5. Review Comments to the Author

Reviewer #1: Mosier and co-authors explored the outcomes of patients affected by acute hypoxemic respiratory failure treated with NIRS (noninvasive respiratory supports) compared to patients initially treated with IMV (invasive mechanical ventilation). They found that initial treatment with NIRS is not associated with a reduced in-hospital death when compared to IMV, but NIRS treatment is associated with a higher likelihood of earlier hospital discharge.

The authors should be commended for the work they have performed. However, I believe that there is a major issue to resolve:

It is not clear if cases analyzed are “de novo acute hypoxemic respiratory failure”, exacerbations or a mix of cases. In the introduction section authors correctly referred to the use of NIRS in acute exacerbation of COPD or acute cardiogenic pulmonary oedema, and in “de novo” hypoxemic respiratory failure (lines 94-99, page 3). However, having a look at data presented in table 1, it comes out that the median pH of the included population is 7.31 (IMV) and 7.33 (NIV). This potentially correlates with the level of CO2 found (46 vs 53). Taking into account that bicarbonate levels were pretty normal or even higher in the noninvasive group, this means that most of the patients were affected already by a chronic respiratory condition. Indeed, 78% and 16% in IMV and 85% and 18% in noninvasive group were respectively COPD or obese. Surprisingly, only 0% was affected by acute respiratory conditions as influenza, and no ARDS patients were enlisted in the retrieved data.

In conclusion, it seems that there is no distinction between de novo and acute exacerbations of chronic condition, and that the majority of this cohort of patients analyzed is affected by exacerbation of chronic condition. I believe that this distinction and a subanalysis of specific populations should also be considered (obesity, COPD, etc).

Response: We thank the reviewer for the thoughtful comments. We narrowed our inclusion criteria based on admission codes that are consistent with acute hypoxemic respiratory failure (AHRF). The reason we chose admission diagnosis rather than discharge or final diagnosis is to better characterize outcomes based on the diagnoses that the clinicians thought they were treating at the time rather than a biased look at a narrower slice of those patients. It does appear that we were successful in narrowing the included population to AHRF based on the median and ranges of both PaO2 and PaO2/FiO2 ratio. We agree that the slightly elevated median PaCO2 correlates well with the slightly acidemic median pH, we disagree with the reviewer’s conclusion that most of the NIRS cohort has a chronic condition for a few reasons. Firstly, as the reviewer points out, the serum bicarbonate is generally normal and based on the H-H equation, using the median numbers, they both calculate as acute respiratory acidoses. Second, the relatively normal serum bicarb and H-H results argue against a chronic condition as the primary cause. Third, there is significant overlap in the IQRs, and the minor differences are accounted for by the use of inverse probability of treatment weighting, thus making broad comparisons about the minor differences in median values is unwarranted. Lastly, given the severity of hypoxemia based on the PaO2 and PF ratio, the degree of mild hypercapnia (based on median values) is certainly within the range of an acute dead space commonly seen in AHRF patients. 

There is a significant number of patients in each cohort, as the reviewer points out, that had comorbid obesity and/or COPD. But these diagnoses are comorbidities and not the primary cause of respiratory failure. This is consistent with data from other studies by Dr. Mosier and collaborators. In a study recently accepted by Critical Care Explorations, a retrospective study in ED patients at the University of Michigan that were treated with noninvasive respiratory support for acute hypoxemic respiratory failure (https://www.medrxiv.org/content/10.1101/2023.09.26.23296167v1) showed that 2/3rds of patients had comorbid COPD and half of patients had a CO2 above expected based on the Winter’s formula, consistent with acute dead space. In the other study, a pilot study conducted at the University of Arizona in preparation for a clinical trial, half of patients with acute hypoxemic respiratory failure had comorbid COPD, and clinicians commonly misattributed (about ½ the time) the admission diagnoses to decompensated COPD or heart failure rather than pneumonia. Thus, extrapolating those findings to this study, since we used admission diagnoses, it may be more likely that we excluded patients with AHRF as the primary cause rather than included patients with decompensated chronic disease as the primary cause. 

Regarding the influenza and ARDS question, we agree these numbers are surprisingly low but we think they are misleading. The reason they may be misleading is that we used admission diagnosis, and influenza results may have been pending at the time the admission diagnosis was entered. Similarly with ARDS, an ARDS diagnosis is commonly a discharge diagnosis, uncommonly an admission diagnosis, and during the timeframe of this study was an impossible diagnosis for any patient in the noninvasive cohort. 

Taken together, the findings of mild respiratory acidoses and a high incidence of comorbid disease is not surprising in patients with acute hypoxemic respiratory failure. However, the reviewer offers a good suggestion about a subgroup analysis. We have performed a secondary subgroup analysis on the primary outcome only, excluding anyone with a comorbidity for COPD or a BMI >35. We found that, while the hazard ratio changed directions, it was still non-significant after excluding these populations and there was no interaction with time. (Figures shown in attached word document of response letter)

In these cumulative incidence curve estimations, every instance suggested that the probability of death was higher with NIRS than with mechanical ventilation. These findings are merely hypothesis generating, but suggestive of a signal towards potential harm in a more heterogeneous patient population. 

(see figures in attached word document of response letter)

We have revised the manuscript to include some of this information while being mindful of word count. 

Reviewer #2: This is an interesting retrospective comparison of in-hospital mortality and the discharge alive of patients with acute respiratory failure treated initially either with noninvasive respiratory support or with invasive mechanical ventilation and deserves to be published. Noninvasive respiratory was associated with more discharges alive, but no reduction in in-hospital mortality at the expense of the high-flow nasal oxygen (HFNO) therapy group. Due to the retrospective nature of the study, treatment with NHFO was problematic and inadequate in relation to the need for respiratory support. Thus, the results do not reflect the actual effectiveness of NHO therapy. It is true, however, that the data comes from real life, even from the pre-Covid-19 period. In addition to the benefits of HFNO in preventing intubation, prior studies have already demonstated the deleterous effect with inadequate use of HFNO. Meanwhile, the knowledge and the way of using NHFO has improved. The authors have already discussed in details the limitations of the study. One of the important issues is the imbalance of the groups. HFNO was the ceiling treatment for terminal-stage diseased patients irrespective of the severity of respiratory failure. Thus, the inclusion criteria for NIPPV group and HFNO group differ significantly as does the outcome.

The authors should address the perspectives of the study in discussion: When is it safe to start HFNO in acute respiratory failure? When to escalate from HFNO to NIPPV or to MV? Is a stepwise approach the right one?

Response: 

We thank the reviewer for the comments and suggestion. We are unsure what is meant by the following: “One of the important issues is the imbalance of the groups. HFNO was the ceiling treatment for terminal-stage diseased patients irrespective of the severity of respiratory failure. Thus, the inclusion criteria for NIPPV group and HFNO group differ significantly as does the outcome.” 

What we interpret this to mean is that nasal high flow is the maximal therapy in end-stage disease patients with do-not-intubate orders, but there is no indication that is any more true in these data than NIPPV being the maximal therapy. We have elaborated the difficulties regarding end-of-life care in the limitations already. Any further elaboration between NIPPV and nasal high flow, which is only a secondary analysis in this study, would be purely speculation given the relatively low numbers of nasal high flow patients after restricting by admission diagnosis codes, and lack of time series information regarding outcomes with crossovers. We made a general comment in this revision in an attempt to address the reviewer’s request with a reference to a recent article by Dr. Mosier that directly addresses this concern, but we feel going any further would be inappropriate in this manuscript.

---

## [Decision Letter · Decision Letter 1]

12 Jul 2024

Noninvasive vs invasive respiratory support for patients with acute hypoxemic respiratory failure

PONE-D-24-02817R1

Dear Dr. Mosier,

We’re pleased to inform you that your manuscript has been judged scientifically suitable for publication and will be formally accepted for publication once it meets all outstanding technical requirements.

Kind regards,

Stefan Grosek, Ph.D., M.D.,

Academic Editor

PLOS ONE

Additional Editor Comments (optional):

Reviewers' comments:

Reviewer's Responses to Questions

**Comments to the Author**

1. If the authors have adequately addressed your comments raised in a previous round of review and you feel that this manuscript is now acceptable for publication, you may indicate that here to bypass the “Comments to the Author” section, enter your conflict of interest statement in the “Confidential to Editor” section, and submit your "Accept" recommendation.

Reviewer #1: All comments have been addressed

Reviewer #2: All comments have been addressed

2. Is the manuscript technically sound, and do the data support the conclusions?

Reviewer #1: Yes

Reviewer #2: Yes

3. Has the statistical analysis been performed appropriately and rigorously? 

Reviewer #1: Yes

Reviewer #2: Yes

4. Have the authors made all data underlying the findings in their manuscript fully available?

Reviewer #1: Yes

Reviewer #2: Yes

5. Is the manuscript presented in an intelligible fashion and written in standard English?

Reviewer #1: Yes

Reviewer #2: Yes

6. Review Comments to the Author

Reviewer #1: (No Response)

Reviewer #2: All the questions were adequately answered. There are no concerns regarding publication ethics and research ethics

7. PLOS authors have the option to publish the peer review history of their article (what does this mean?). If published, this will include your full peer review and any attached files.

Reviewer #1: No

Reviewer #2: **Yes: **Andreja Sinkovič

---

## [Editor Report · Acceptance letter]

25 Jul 2024

PONE-D-24-02817R1 

PLOS ONE

Dear Dr. Mosier, 

I'm pleased to inform you that your manuscript has been deemed suitable for publication in PLOS ONE. Congratulations! Your manuscript is now being handed over to our production team.

Kind regards, 

on behalf of

Professor Stefan Grosek 

Academic Editor

PLOS ONE